# Capsaicin-Loaded Chitosan Nanocapsules for wtCFTR-mRNA Delivery to a Cystic Fibrosis Cell Line

**DOI:** 10.3390/biomedicines8090364

**Published:** 2020-09-20

**Authors:** A. Katharina Kolonko, Janes Efing, Yadira González-Espinosa, Nadine Bangel-Ruland, Willy van Driessche, Francisco M. Goycoolea, Wolf-Michael Weber

**Affiliations:** 1Institute of Animal Physiology, University of Muenster, Schlossplatz 8, 48143 Muenster, Germany; j_efin04@wwu.de (J.E.); n.br@wwu.de (N.B.-R.); wmw@wwu.de (W.-M.W.); 2School of Food Science and Nutrition, University of Leeds, Leeds LS2 9JT, UK; Y.GonzalezEspinosa@leeds.ac.uk (Y.G.-E.); F.M.Goycoolea@leeds.ac.uk (F.M.G.); 3EP-Devices, Tervuursesteenweg 154, 3060 Bertem, Belgium; ep.devices@telenet.be

**Keywords:** CFTR, ENaC, chitosan, mRNA therapy, nanocapsules, cystic fibrosis, capsaicin, *transcript therapy*

## Abstract

Cystic fibrosis (CF), a lethal hereditary disease caused by mutations in the *cystic fibrosis transmembrane conductance regulator* (CFTR) gene coding for an epithelial chloride channel, is characterized by an imbalanced homeostasis of ion and water transports in secretory epithelia. As the disease is single-gene based, *transcript therapy* using therapeutic mRNA is a promising concept of treatment in order to correct many aspects of the fatal pathology on a cellular level. Hence, we developed chitosan nanocapsules surface-loaded with wtCFTR-mRNA to restore CFTR function. Furthermore, we loaded the nanocapsules with capsaicin, aiming to enhance the overall efficiency of *transcript therapy* by reducing sodium hyperabsorption by the epithelial sodium channel (ENaC). Dynamic light scattering with non-invasive back scattering (DLS-NIBS) revealed nanocapsules with an average hydrodynamic diameter of ~200 nm and a Zeta potential of ~+60 mV. The results of DLS-NIBS measurements were confirmed by asymmetric flow field-flow fractionation (AF4) with multidetection, while transmission electron microscopy (TEM) images confirmed the spherical morphology and size range. After stability measurements showed that the nanocapsules were highly stable in cell culture transfection medium, and cytotoxicity was ruled out, transfection experiments were performed with the CF cell line CFBE41o-. Finally, transepithelial measurements with a new state-of-the-art Ussing chamber confirmed successfully restored CFTR function in transfected cells. This study demonstrates that CS nanocapsules as a natural and non-toxic delivery system for mRNA to target cells could effectively replace risky vectors for gene delivery. The nanocapsules are not only suitable as a *transcript therapy* for treatment of CF, but open aspiring possibilities for safe gene delivery in general.

## 1. Introduction

Messenger RNA (mRNA) is a single-stranded nucleic acid, which transfers genetic information to the ribosome, where protein synthesis takes place. The use of mRNA for therapeutic purposes, such as the so-called *transcript therapy*, has been tested since the early 1990s and has undergone tremendous developments since then [1]. In fact, mRNA displays many advantages for gene delivery compared to DNA. For example, there is no need for nuclear localization facilitating the delivery to the target, namely the cytosol, and the risk of insertional mutagenesis, followed by oncogenic effects, is being avoided. Furthermore, mRNA is smaller than DNA, providing easier transport into the cell; it also exhibits reduced immunogenicity and is efficient at any point of the cell cycle [2]. Currently, three clinical trials are testing the efficiency of mRNA-based therapies targeting different proteins, including the cystic fibrosis transmembrane conductance regulator (CFTR) via lipid nanoparticle inhalation for the treatment of cystic fibrosis (CF) [3].

CF, being the most common lethal genetic disorder in Caucasians, is caused by mutations in the CFTR gene encoding for a cAMP-dependent chloride and bicarbonate channel mainly expressed in the apical membrane of secretory epithelial cells. The defect chloride secretion through CFTR is associated with sodium hyperabsorption via the amiloride-sensitive epithelial sodium channel (ENaC). The classical clinical picture caused by the imbalanced homeostasis of ion and water transport is mainly characterized by fat maldigestion due to pancreatic insufficiency [4,5] and chronic obstructive lung disease with bacterial colonization by microorganisms such as *Pseudomonas aeruginosa* and *Staphylococcus aureus* [6]. Thanks to improved treatment and early diagnosis, the median survival for patients born in 2000 is expected to be in excess of 50 years [7]. Even though innovative treatment strategies, such as personalized medicine using CFTR modulators, have improved the quality of life, as well as the life expectancy of patients significantly [8,9], a causative treatment for CF still needs to be found in order to treat all patients, regardless of their genotype.

In our group, we have been working with wtCFTR-mRNA to improve chloride secretion of epithelial cells constantly optimizing delivery methods [10,11,12]. The most promising delivery systems of nucleic acids to epithelial cells proved to be self-assembled polyelectrolyte chitosan (CS)-based nanocomplexes due to their low cytotoxicity and high transfection efficiency [12,13,14]. CS itself, a family of pseudonatural polysaccharides composed of randomly distributed β-(1-4)-linked D-glucosamine and *N*-acetyl-D-glucosamine units, is a promising candidate for non-viral gene delivery [15]. Because of its positive attributes, such as biocompatibility, mucoadhesiveness, low cytotoxicity, and low immunogenicity, CS has already been used as a carrier for various drugs in multiple studies [16,17,18]. However, our matrix-type nanosystems displayed poor stability in transfection medium, possibly reducing transfection efficacy. Therefore, we aimed at developing a new delivery system with improved stability and higher transfection efficiency.

We prepared CS-lecithin oil-core nanocapsules with the solvent displacement technique according to the principle of spontaneous emulsification [19,20]. Briefly, these nanocapsules consist of an oil-core lined with lecithin. The hydrophilic head groups of the phospholipidic surfactant face to the outside, enabling the phospholipids to interact with oppositely charged CS via electrostatic binding forming a core-shell colloidal nanocapsule. For transfection purposes, we loaded wtCFTR-mRNA to the surface of the nanosystems, using a layer-by-layer electrostatic assembly principle, thus forming a polyelectrolyte complex shell. Furthermore, we loaded capsaicin in the oil-core of the nanocapsules with a view to potentially increase transfection efficiency, as the compound has been found to reversibly open tight junctions [21,22] and to have mucus thinning properties [23,24], which can be advantageous in the CF lung.

Capsaicin, the main pungent alkaloid found in members of the *Capsicum* family, e.g., the chili pepper, is the agonist to the transient receptor potential vanilloid 1 (TRPV1), a ligand-gated, calcium-permeable cation channel mainly expressed in cells of the nervous system [25,26]. Capsaicin has been known for its wide range of therapeutic properties since ancient times [27]. The current main therapeutic application is as an analgesic in topical creams and dermal patches against neuropathic pain [28,29]. However, many other therapeutic benefits have been described [30]. Recent studies have documented that capsaicin has a direct effect on urinary sodium excretion by inhibiting ENaC function and thereby affecting renal sodium reabsorption [31]. We have, therefore, hypothesized that bundling capsaicin in a nanocapsules formulation, including electrostatically bound CS and wtCFTR-mRNA, would not only lead to an increased transfection efficiency but, as a secondary effect, also to a decreased ENaC activity; thus, this approach may result in a double-tracked strategy to target CFTR and ENaC simultaneously.

In this study, we aimed at developing stable CS nanocapsules incorporating capsaicin to increase transfection efficiency of wtCFTR-mRNA in a CF cell line. Our results indicate successfully restored CFTR function after treatment with the CS nanocapsules, demonstrating the aspiring possibilities of a non-viral *transcript therapy* as a treatment for CF.

## 2. Experimental Section

### 2.1. mRNA Synthesis

For synthesis of wtCFTR-mRNA via in vitro transcription (IVT), the construct pSTI-A120/hCFTR-cDNA provided by C. Rudolph (Ludwig Maximilian University of Munich, Munich, Germany) was used. The construct pALTER-A120/hΔF508-CFTR provided by K. Kunzelmann (University of Regensburg, Regensburg, Germany) was used for synthesis of ΔF508-CFTR-mRNA. The mRNA synthesis was performed as described previously [10]. Briefly, the linearized plasmid was precipitated with ethanol/sodium acetate after extraction with phenol/chloroform. The IVT reaction was performed with the mMESSAGE mMACHINE™ T7 Transcription Kit (Invitrogen, Carlsbad, CA, USA). For purification of the reaction, the RNeasy^®^ Plus Mini Kit (Qiagen, Hilden, Germany) was used. Finally, the mRNA was precipitated with ethanol/ammonium acetate and dissolved in RNase-free water. The concentration of the product was determined by absorbance measurement at 260 nm, using a microvolume spectrophotometer (NanoDrop 1000, Peqlab Biotechnologie GmbH, Erlangen, Germany), while the integrity and size distribution was determined by agarose-formaldehyde gel electrophoresis. The mRNA was stored at −20 °C, in nuclease-free water.

### 2.2. Chitosan-Lecithin Oil-Core Nanocapsules

The CS was purchased from HMC+ GmbH (Halle/Saale, Germany) as an ultrapure biomedical-grade sample (Heppe 70/5; Batch-No. 212-170614-01; DA = 17%; Mw = 29.3%). It was dissolved in water with 5% stoichiometric excess of 5 M HCl.

The nanocapsules (NC) were prepared as described previously, with slight modifications [22]. Briefly, 400 µL ethanolic lecithin solution (100 mg/mL) and 530 µL capsaicin stock solution (24 mg/mL in ethanol) were mixed and supplemented with 125 µL Miglyol^®^ 812 N and 9.5 mL ethanol. Under constant stirring, this organic solution was then poured into 20 mL aqueous CS solution (0.5 mg/mL). Concentration of the milky mixture was performed at 40 °C, using a rotavapor (Büchi R-210, Büchi Labortechnik GmbH, Essen, Germany). During the process, the pressure was slowly decreased to 0 mbar, until 3.5–4.0 mL remained. To yield a final CS concentration of 2.5 mg/mL and a final capsaicin concentration of 10 mM, the volume was topped up to 4.0 mL with water. Unloaded nanocapsules were prepared similarly by using ethanol instead of the capsaicin solution.

The nanocapsules were loaded with mRNA by carefully mixing 15 µL of nanocapsules with different volumes of wtCFTR-mRNA (0.053 µg/µL), to reach varying charge ratios (Table 1). The nanocapsules were incubated for 30 min, at room temperature (RT), to allow complexation of the mRNA and the CS.

### 2.3. Determination of Size Distribution and Zeta Potential

Dynamic light scattering with non-invasive back scattering (DLS-NIBS) with a measurement angle of 173° was performed to measure the size distribution of the nanocapsules [32]. Their zeta potential (*ζ*) was determined by applying the Henry equation, using the Smolouchowski approximation after exerting laser Doppler microelectrophoresis and phase analysis light scattering (M3-PALS) [33]. For measurements, the nanocapsules were diluted in water (1:67) and filled into a folded capillary zeta potential cell (Model DTS1070; Malvern Panalytical Ltd., Worcestershire, UK). Measurements were performed at 25 °C with a Zetasizer Nano ZS 6300 instrument (Malvern Panalytical Ltd., Worcestershire, UK). Data were acquired with 4 measurements each, with runs set automatically.

### 2.4. Stability Measurements

The size distribution of the nanocapsules was measured at different time points by DLS-NIBS, with a measurement angel of 173°, in order to determine their stability in transfection medium. The nanocapsules were prepared at RT and diluted in transfection medium (1:67; Opti-MEM™ (Thermo Fisher Scientific, Waltham, MA, USA) supplemented with 20 mM HEPES and 270 mM mannitol). Subsequently, nanocapsules were incubated at 37 °C in between measurements. Measurements were performed after 0, 4, and 24 h, at 37 °C, using a low volume disposable cuvette (Sarstedt AG & Co, Hemer, Germany) with a Zetasizer Nano ZS 6300 instrument (Malvern Panalytical Ltd., Worcestershire, UK). Data were acquired with 4 measurements each, with runs set automatically.

### 2.5. Gel Retardation Assay

To test the binding efficiency of nucleic acids to CS, a gel retardation assay was carried out. For preparation of the samples, 5 µL wtCFTR-mRNA (0.053 µg/µL) was mixed with 5 µL nanocapsules (blank and loaded) and incubated for 30 min at RT, or were incubated in 20 µL Opti-MEM™, at 37 °C, for 24 h. After addition of 2 × RNA loading dye (Thermo Fisher Scientific, Waltham, MA, USA), the nanosystems were loaded onto a 1% agarose-formaldehyde gel in RNA running buffer (0.1 M Mops, 50 mM sodium acetate, 5 mM EDTA; pH 7) and electrophoresed at 128 V for 45 min. Finally, the nucleic acid bands were visualized in a BioDocAnalyze System (Analytik Jena, Jena, Germany).

### 2.6. Asymmetric Flow Field-Flow Fractionation

Asymmetric flow field-flow fractionation (AF4) measurements [34] were performed on an AF2000 Multiflow system from Postnova Analytics GmbH (Landsberg am Lech, Germany). The system was set to be operated in AF4 mode and was coupled with an online 21 angle multi-angle light scattering detector MALS (PN3621), a refractive index detector RI (PN3150), and a dual wavelength UV detector (PN3211) set at 280 and 220 nm. A DLS-NIBS detector (Zetasizer Nano ZS, Malvern Panalytical Ltd., Worcestershire, UK) was fitted online for determination of the hydrodynamic radius, using a light path quartz flow cell (ZEN0023) and 173° backscattered angle. The AF4 system was equipped with an analytical asymmetric AF4 cartridge design channel (Postnova Z-AF4-CHA-611, Postnova Analytics GmbH, Landsberg am Lech, Germany), using a 350 µm spacer. A thermostat (PN4020) set at 30 °C for all experiments controlled the temperature of the channel, and a membrane with a 10 kDa cutoff made of regenerated cellulose was used (Z-AF4-MEM-612-10KD, Postnova Analytics, Landsberg am Lech, Germany). In order to minimize the interactions between the positively charged nanocapsules and the residual negatively charged groups of the membrane, an acetate buffer solution (0.18 M acetic acid, 0.02 M sodium acetate; pH 3.7) was used as a carrier liquid, leading to the surface of the membrane being charged positively (*ζ* ~+10 mV). This favors the polymer elution generating repulsive forces between both the positively charged CS-covered nanocapsules and cellulose membrane [35].

For experiments, the nanocapsules were diluted in the acetate buffer solution (4:1), and for each run, 2 µL of sample was injected into the system. The elution was carried out by a cross-flow programmed with a time delay exponential decay. The sample was injected at 0.2 mL/min and then focused at a rate of 1.3 mL/min, for 3 min, with the cross-flow set at 1 mL/min. After the focusing period and a transition period of 0.2 min, the profile of the cross-flow was gradually decreased over one hour, with a series of consecutive steps. (a) The cross-flow was kept constant at 1 mL/min for 0.2 min. (b) The cross-flow was decreased at an exponent decay from 0.2 to 0.1 mL/min over 40 min. (c) Finally, the cross-flow was kept constant at 0.1 mL/min for 20 min. During the entire process, including the focus step, the detector flow was maintained at 0.5 mL/min. Data collection and analysis were achieved with NovaFFF software version 2.0.9.9. (Postnova Analytics GmbH, Landsberg am Lech, Germany), and data were best fitted to random coil model.

### 2.7. Transmission Electron Microscopy

The ultrastructure of the nanocapsules was investigated by using transmission electron microscopy (TEM). Equal amounts of samples were mixed with uranyl acetate solution (negative staining, 1% *w*/*v*). Samples (8 µL) were placed onto a copper grid covered with Formvar^®^ film, and the excess of liquid was removed with the aid of a filter paper. The analyses were performed by using JEM-1400 TEM (JEOL, Peabody, MA, USA), operated at 100 kV, and captured on AMT 1K CCD, using the software AMTV602 (Advanced Microscopy Techniques, Woburn, MA, USA).

### 2.8. Cell Culture

Dr. Dieter Gruenert (Department of Otolaryngology—Head and Neck Surgery, University of California, San Francisco, CA, USA) provided 16HBE14o- and CFBE41o- cells. Cells were cultivated in T-75 flasks on 1% fibronectin in Minimal Essential Medium Earle’s supplemented with 10% fetal calf serum, 2 mM L-glutamine, and 100 U/mL penicillin/streptomycin, at 37 °C, with 5% CO_2_ and 95% rH. For Ussing chamber experiments, 1.15 × 10^5^ cells were seeded on 1% fibronectin coated Costar Transwell^®^permeable filters (*Ø* = 6.5 mm; REF 3470; Corning Inc., Lowell, MA, USA). The medium was changed every 2–3 days, and experiments were conducted after 7–10 days.

### 2.9. MTT Assay

Cytotoxicity of the nanocapsules was analyzed with a 3-(4,5-dimethylthiazol-2-yl)-2,5-diphenyltetrazolium bromide (MTT) assay as described previously [14]. Briefly, CFBE41o- and 16HBE14o- cells were seeded in a 96-well microtiter plate (1 × 10^4^ cells per well) and incubated for 24 h at 37 °C, at 5% CO_2_ and 95% rH. Before application of the samples, the cells were washed twice with serum-free cell culture medium. The cells were then incubated with the samples, under previous conditions. After 24 h, the samples were removed and replaced by 100 μL serum-free cell culture medium and 25 μL MTT solution (5 mg/mL in PBS), followed by another incubation of 4 h, to allow formation of the purple formazan salt. The medium was removed, and the formazan was dissolved with 100 µL dimethyl sulfoxide for 30 min, at 37 °C. Finally, the absorbance was measured at *λ* = 570 nm in a microplate reader (EZ Read 400, Biochrom GmbH, Berlin, Germany), and relative viability was calculated by dividing individual viabilities by the mean of the negative control (serum-free cell culture medium). Then 1% Triton^®^ X-100 was used as a positive control.

### 2.10. Transfection

Twenty-four hours before transfection, the medium of the CFBE41o- cells was replaced with fresh antibiotic-free medium. CFBE41o- cells were transfected with 2.4 µg/cm^2^ wtCFTR-mRNA or ΔF508-CFTR-mRNA, respectively, using 1 µL of either blank or capsaicin-loaded nanocapsules per Costar Transwell^®^permeable filter. Briefly, nanocapsules were mixed with the desired amount of mRNA (0.3 µg/µL) and topped up with water, to a volume of 50 µL per filter. The formulations were incubated at RT for 30 min to allow loading of the mRNA to the CS nanocapsules. After incubation, the nanocapsules were mixed with Opti-MEM™ to reach a final volume of 100 μL per filter and incubated for another 5 min at RT. Finally, the nanocapsules were added to the cells on the apical side of the Costar Transwell^®^permeable filter, together with 200 µL of fresh cell culture medium, to reach a capsaicin concentration of 33 µM. Cells were incubated for 24 h at 37 °C, at 5% CO_2_ and 95% rH, before experiments were conducted.

### 2.11. Transepithelial Measurements

Transepithelial short-circuit current (I_sc_) through monolayers cultured on Costar Transwell^®^permeable filters was recorded with a transepithelial current clamp and a microcontroller based data acquisition system (EP-Devices, Bertem, Belgium). I_sc_ was calculated as ratio of open circuit potential (PD) by transepithelial resistance (R_t_). R_t_ was recorded from the voltage response to a 1 Hz sine wave current stimulus. The data acquisition system was connected to an USB port of a PC to display and store data. The instrument was built to simultaneously record R_t_ and PD from four monolayers. The membrane area of the Costar Transwell^®^permeable filters is 0.33 cm^2^. Four filters were mounted in four compartments in a Lucite holder. These compartments constituted the basolateral bathing side and were filled with 1 mL of physiological medium. The volume of solution at the apical side was 250 µL. Recording of R_t_ was done with a four electrode arrangement, consisting of voltage sensing and current sending electrodes. Voltage and current electrodes were Ag/AgCl pellets mounted in a holder, for connection to the data acquisition system. The four-channel manifold could be accurately placed on the Lucite holder, and electrodes were positioned in the apical and basolateral compartment. To add solutions, electrodes were moved to reference compartments in the Lucite filter holder, where voltage electrode offsets were recorded. The Lucite holder was mounted in a heating block with a temperature controller powered with 24 Volt DC supply. Temperature control was set at 37 °C, and sensing was done with a P_t_ resistance probe inserted in the Lucite holder close to the filter compartments. Basolateral bathing solutions were bubbled with air.

First, sodium current mediated by ENaC was evaluated. After adapting the cells to the physiological Ringer solution (130 mM NaCl, 5 mM KCl, 2 mM MgCl_2_, 1 mM CaCl_2_, 5 mM glucose, and 10 mM HEPES; pH 7.3; 37 °C), sodium absorption through ENaC was blocked by apical application of amiloride (10 μM; Sigma-Aldrich, St. Louis, MO, USA). To determine the overall sodium absorption of the cells, sodium-free Ringer solution (130 mM tetramethylammonium chloride instead of NaCl) was applied to the apical side afterward. To determine CFTR activity, ENaC was blocked again by apical application of amiloride before an activation cocktail consisting of the membrane permeable cAMP analogue 8-[4-chlorophenylthio (CTP)]-cAMP (100 µM; Biolog, Bremen, Germany) and IBMX (1 mM; Sigma-Aldrich, St. Louis, MO, USA) was applied to the basolateral side of the monolayer. The chloride current was blocked, using the specific CFTR channel blocker CFTRinh172 (10 µM in amiloride solution; Tocris Cookson, Ellisville, MO, USA).

### 2.12. Statistical Analysis

GraphPad Prism^®^ Version 6.01 (GraphPad Software Inc., La Jolla, CA, USA) was used for the statistical analysis. If not stated otherwise, the arithmetic mean values ± standard deviations (SD) of at least three independent experiments were determined. The data of the cell culture experiments are expressed as the arithmetic mean values ± standard error of the mean (SEM). The non-parametric Kruskal–Wallis test was performed to assess the significant differences of the data. Differences were considered statistically significant when *p* ≤ 0.05 (*), *p* ≤ 0.01 (**), *p* ≤ 0.001 (***), and *p* ≤ 0.0001 (****).

## 3. Results

### 3.1. Physicochemical Characterization of Nanocapsules

The physicochemical properties of blank (NC Blank), as well as capsaicin-loaded (NC CAP) nanocapsules, either naked or co-loaded with wtCFTR-mRNA at P/N charge ratio 75, were determined with DLS-NIBS measurements and from their electrophoretic mobility. This specific P/N charge ratio was chosen based on the transfection procedure as at P/N 75 the amount of wtCFTR-mRNA, as well as capsaicin are ideal for transfection of CFBE41o- cells. The Z-average hydrodynamic diameter of the nanocapsules increased significantly after loading of capsaicin (Figure 1a). This outcome can also be noticed in the intensity plot as the peak of the capsaicin-loaded nanocapsules shifts slightly to the right. While the size of the nanocapsules increased, the polydispersity index (PDI) decreased significantly for capsaicin-loaded nanocapsules (Figure 1b). In general, the nanocapsules appear to be highly monodisperse as seen in the low PDI, as well as the narrow size distribution showing only one monomodal population (Figure 1c). The increase in size runs in parallel with the nanocapsules’ zeta potential, which also shows an increase after loading of capsaicin (Figure 1d). For all attributes, no effect was noticeable after surface-loading the nanocapsules with wtCFTR-mRNA. The physicochemical properties of the nanocapsules are summarized in Appendix A
Appendix A.

Blank and capsaicin-loaded nanocapsules were also successfully characterized by AF4, as shown by an increasing hydrodynamic radius (R_h_) with increasing elution time in the elugrams shown in Figure 2. The gyration radius (R_g_) also increased with advancing elution. Capsaicin-loaded nanocapsules eluted slightly later than did blank nanocapsules, giving an indication of the greater size of the loaded nanocapsules and confirmed by the DLS-NIBS online measurements. The shift in elution is also observed in the comparison of the R_g_/R_h_ ratios (shape factors) presented in Figure 2e. Even though the ratios of both nanocapsules seem to be comparable, capsaicin-loaded nanocapsules eluted later than blank nanocapsules. The size distribution parameters measured by AF4 are reported in Table 2. As seen in Figure 2e, blank and loaded nanocapsules had similar shape factors, indicating identical shapes of both.

For further analysis and visualization, TEM images were taken of blank and capsaicin-loaded nanocapsules (Figure 3). The images show that capsaicin-loaded nanocapsules are larger than blank nanocapsules, confirming the DLS-NIBS and AF4 measurements. As seen in the images, the size of blank nanocapsules varies between 100 and 150 nm, while capsaicin-loaded capsules show a diameter of 200 to 250 nm, coinciding with DLS-NIBS measurements (137.5 ± 8.2 vs. 216.6 ± 35.3 nm). The shape of the nanocapsules is also visible in the TEM images. As it seems, the nanocapsules are not homogeneous or entirely spherical. On the contrary, some of the depicted nanocapsules appear to be slightly elliptical.

In order to test the effect of the nucleic acid loaded to the surface of the nanocapsules, their physicochemical properties were characterized again. Capsaicin-loaded nanocapsules were co-loaded with wtCFTR-mRNA, in a broad range of P/N charge ratios (5, 10, 50, 75, and 100), followed by DLS-NIBS and electrophoretic mobility measurements, as described previously. The nucleic acid affected neither the Z-average hydrodynamic diameter nor the PDI of the nanocapsules. As seen in Figure 4a, the size of the nanocapsules varied between 196 and 205 nm for all P/N charge ratios. The PDI of the nanocapsules remained at a very low level around ~0.1 (Figure 4b). Analysis of size and PDI measurements were confirmed by the relative intensity of scattered light, showing no shifts between the different P/N charge ratios (Figure 4c).

While size and PDI remained stable, the zeta potential of the nanocapsules increased with increasing P/N charge ratio (+59 to +63 mV; Figure 4d). This effect indicates that the addition of mRNA leads to modification of the nanocapsules’ electrophoretic mobility, suggesting successful surface-loading of the wtCFTR-mRNA to the CS nanocapsules.

To verify the results of the zeta potential measurements arguing strongly in favor of the electrostatic binding of the mRNA at the surface of the nanocapsules, a gel retardation assay was carried out to investigate the binding efficiency of wtCFTR-mRNA and nanocapsules (Figure 5). The naked wtCFTR-mRNA ran through the gel, forming a specific band around 4000 bases, corresponding to its size (4563 bases), while both types of nanocapsules were retained in the gel pockets because of their positive charge. No specific bands were visible in the lanes of the gene-loaded nanocapsules in water or in transfection medium (Opti-MEM™), in good keeping with the zeta potential measurements and suggesting the successful surface-loading of wtCFTR-mRNA onto the CS nanocapsules and high stability in transfection medium. The original image of the gel retardation assay can be found in the Appendix A
Appendix A.

### 3.2. Stability of Nanocapsules

The stability of the nanocapsules in transfection medium was examined before cell culture experiments were conducted. Capsaicin-loaded nanocapsules were incubated in Opti-MEM™ supplemented with HEPES and mannitol at 37 °C for 24 h, either naked or surface-loaded with wtCFTR-mRNA at P/N charge ratio 75. The size of the nanocapsules was determined by DLS-NIBS at different time points (0, 4, and 24 h). Both naked and surface-loaded nanocapsules remained extremely stable over time, as shown in Figure 6. The PDI of the nanocapsules stayed at low values between 0.07 and 0.09, and Z-average hydrodynamic diameter remained stable around 200 nm throughout 24 h, showing no shifts in the monomodal peaks of the intensity plots.

### 3.3. Cell Culture Experiments with Nanocapsules

The cytotoxicity of the nanocapsules was evaluated by an MTT assay (Figure 7). CFBE41o- cells from a human bronchial CF cell line, homozygous for the ΔF508 mutation, were incubated with blank and capsaicin-loaded nanocapsules at varying concentrations, for 24 h, before the cell viability was assessed. Neither blank nor capsaicin-loaded nanocapsules showed cytotoxic effects at low concentrations of 0.33% corresponding to 33 µM of capsaicin. Covering the capsules with mRNA did not alter the effect. However, increasing concentrations of nanocapsules also increased their cytotoxic effects, showing reduced cell viability starting with nanocapsule concentrations of 1%. While capsaicin-loaded nanocapsules only decreased the cell viability significantly at concentrations of 5% (500 µM capsaicin; 40.9 ± 4.0%), blank nanocapsules already showed significantly cytotoxic effects at 2.5% (33.8 ± 2.8%). Blank nanocapsules at concentrations of 5% significantly decreased the cell viability to the level of the positive control (19.4 ± 1.9% vs. 21.9 ± 1.2%). The cell viability of the healthy equivalent cell line 16HBE14o- was affected similarly, as shown in Appendix A
Appendix A.

Based on the MTT assay, CFBE41o- cells were transfected with 2.4 µg/cm^2^ wtCFTR-mRNA or ΔF508-CFTR-mRNA using 0.33% nanocapsules. Then, 24 h after transfection, functional Ussing chamber measurements were carried out in order to evaluate the effect of the transfection. As control, the healthy equivalent bronchial epithelial cell line 16HBE14o- was used.

Transfection with wtCFTR-mRNA, using blank nanocapsules, did not lead to a change in potential difference after application of cAMP, while the cAMP response was increased in cells transfected with capsaicin-loaded nanocapsules (Figure 8a). Statistical evaluation of the short-circuit current (I_sc_) showed that the increase in cAMP-mediated current was significant, indicating successful transfection and CFTR expression (1.5 ± 0.3 μA/cm^2^ vs. 3.3 ± 0.6 μA/cm^2^; Figure 8b). As expected, transfection with ΔF508-CFTR-mRNA, using either blank or capsaicin-loaded nanocapsules, did not lead to a change in cAMP response, corroborating the results achieved by transfection with wtCFTR-mRNA. Furthermore, transepithelial measurements revealed significantly higher cAMP response in healthy control cells, compared to the CF cell line CFBE41o-, indicating higher CFTR activity and thus validating the expected behavior of the cells.

In order to estimate a possible effect of wtCFTR-mRNA transfection on ENaC, the amiloride-sensitive current was evaluated (Figure 8c). After transfection using blank nanocapsules, amiloride-sensitive current increased significantly. Treatment with capsaicin-loaded nanocapsules led to a non-significant increase in amiloride-sensitive current. Transfection with ΔF508-CFTR-mRNA did not alter the amiloride-sensitive current significantly, as compared to untreated CFBE41o- cells, even though the cells treated with capsaicin-loaded nanocapsules displayed a slightly lower amiloride-sensitive current. However, no difference in amiloride-sensitive current was observed between CFBE41o- and healthy control cells 16HBE14o-, thus casting doubt on the validity of the cell lines regarding ENaC activity.

## 4. Discussion

### 4.1. Highly Monodisperse and Positively Charged Nanocapsules Successfully Load wtCFTR-mRNA

In recent years, CS-based nanocapsules have been gaining traction for the delivery of drugs, peptides, or antigens, to name a few [36,37,38,39,40]. This system was originally pioneered by the group of Prof. Maria J. Alonso [41]. The optimization of the composition and preparation conditions were optimized [39], and the role of the variation on the structural characteristics of chitosan (degree of acetylation and molecular weight) on the physical characteristics, colloidal stability, and capsaicin-loading efficiency were also subsequently studied [19,42].

Despite the many studies on CS nanocapsules as drug delivery systems, only very few have reported about the delivery of genetic material [43,44]. Moreover, these nanosystems encapsulate the nucleic acids rather than loading them to their CS-covered surfaces, making the approach of this study highly exceptional.

As CS-based nanocapsules are known for their increased stability and their interaction with mucosal surfaces, making them interesting for transmucosal drug delivery [19,39,45], we aimed at developing a similar system to improve transfection efficacy in the hostile milieu of the CF lung. We obtained CS-covered nanocapsules either blank or loaded with the vanilloid capsaicin by applying the spontaneous emulsification technique, which is also known as solvent displacement [20]. Features such as physical and chemical properties, as well as biological performance (e.g., transfection efficiency) of CS, are dependent on molecular weight and degree of acetylation of the polysaccharide [46]. Therefore, it is crucial to rigorously characterize CS in order to design the most optimal formulation for a specific purpose. To further optimize the suitability of CS-based nanocarriers as delivery systems, it is essential to characterize them for their physicochemical properties and understand them on a molecular level [15,47]. Thus, we characterized the nanocapsules regarding their physicochemical properties. The Z-average hydrodynamic diameter increased around ~50 nm after loading the nanocapsules with capsaicin revealed by DLS-NIBS measurements (Figure 1a). These results indicate successful encapsulation of the vanilloid, as described before [19,22]. In fact, a previous study reported a strong capsaicin-association efficiency (92%) in CS-lecithin oil-core nanocapsules [22], thus further validating successful encapsulation. Measurements of the nanocapsules’ electrophoretic mobility also showed increased zeta potential after encapsulation of capsaicin (Figure 1d). The increased positive charge of the nanocapsules arises from the enlarged surface of the nanosystems exposing more positively charged CS on the outside. In general, the nanocapsules displayed many auspicious attributes, such as small size, low PDI, and high zeta potential (Appendix A
Appendix A).

To further analyze the nanosystems in regard to their size and shape, they were characterized upon fractionation by AF4 with multidetection. As the DLS-NIBS measurements already indicated, the nanocapsules’ AF4 elution profiles revealed that the capsaicin-loaded nanocapsules are bigger than their blank equivalent (Figure 2). Furthermore, the shape factor (R_g_/R_h_) was calculated from the radii obtained from MALS and DLS-NIBS online measurements and, in this case, gives an indication of the morphology of the nanocapsules. The shape factors for both blank and loaded nanocapsules were around 0.69–0.70 (Table 2), a value which is lower than that expected for spheres (0.775). These lower values are characteristic of core–shell structures, indicating that the mass concentrates in the core region [48]. Furthermore, theoretical R_g_/R_h_ ratio values for hard (0.775), homogeneous (0.778), and hollow (1.0) spheres have been reported [49,50]. The lower R_g_/R_h_ values found for the nanocapsules in our study can be explained as a deviation from the standard spherical models. This hypothesis was confirmed by TEM imaging that revealed few spherical shapes but also partially elliptical and vesicle-shaped smaller structures (Figure 3).

After general characterization of the nanocapsules, they were further investigated regarding their physicochemical properties when surface-loaded with wtCFTR-mRNA at varying P/N charge ratios (5, 10, 50, 75, and 100). The Z-average hydrodynamic diameter, as well as the PDI of the nanocapsules, remained stable (Figure 4a–c). This might be explained by the fact that we only used very low amounts of mRNA, which are not enough to result in an increase of thickness of the shell. In contrast, the zeta potential of the nanocapsules increased with increasing P/N charge ratio (Figure 4d). This expected increase demonstrates successful loading of the nucleic acid to the surface of the nanocapsules. With increasing P/N charge ratio, less mRNA is applied to the nanocapsules, and, therefore, fewer negatively charged phosphate groups can take part in the neutralization of the positively charged amine groups of the CS leading to an increase in zeta potential. This phenomenon was described before, in this study and by others [13,51,52,53], validating the result of zeta potential measurements. The gel retardation assay shown in Figure 5 further enforces the successful surface-loading of wtCFTR-mRNA to the nanocapsules, as the nucleic acid was completely retained in the gel pocket by the CS-covered nanosystems. The outcome of the gel retardation assay corroborates previous reports showing overall strong binding efficiency of nucleic acids and CS [13,14,54,55].

The small size below 500 nm, low PDI, and highly positive zeta potential increase their chance of endocytotic uptake because of their interaction with the negatively charged cell membrane [56,57]. These attributes, as well as the successful loading of wtCFTR-mRNA to the surface of the CS-covered nanocapsules, make the system very promising for transfection purposes.

### 4.2. Nanocapsules are Highly Stable in Transfection Medium

Aggregation of nanosystems can significantly alter their fate, such as uptake by cells, as well as their cytotoxicity or biodistribution, to name a few [58]. Therefore, it is indispensable to evaluate their colloidal stability in a cell culture environment in order to guarantee successful transfection and effective translation into clinical applications. Hence, the stability of the nanocapsules in transfection medium was evaluated before cell culture experiments were performed. Naked and surface-loaded nanocapsules were incubated in Opti-MEM™ supplemented with HEPES and mannitol for 24 h at 37 °C. The supplements were chosen based on a recommendation for a CS-based transfection reagent from NovaMatrix^®^ (Sandvika, Norway). The idea behind supplementation of the transfection medium is the stabilization of the nanosystems with short-range repulsive hydration forces, which are produced by accumulation of the reagents close to their hydrophilic surface [13]. Both systems remained stable over time (Figure 6), as reported by previous studies [22,59]. However, the transfection medium supplemented with HEPES and mannitol was reported to display a very high osmolality, around 580 mOsmol/kg, having a shrinking effect on the cells [14]. Based on these data, transfection experiments should be conducted in medium without supplements. Previous studies showed that CS-lecithin oil-core nanocapsules also remained stable in unsupplemented medium, which can be applied to our systems, as well [22,59]. Therefore, transfection experiments were performed, using unsupplemented Opti-MEM™.

### 4.3. wtCFTR-mRNA Transfection Using Capsaicin-Loaded Nanocapsules Restores CFTR Function

The concept of mRNA delivery to cells for therapeutic purposes, referred to as *transcript therapy*, has been on the rise in the last years [2,3,10]. The use of mRNA instead of DNA, like in the classical definition of gene therapy, displays many advantages, such as higher translation efficacy, decreased immunogenicity, and increased overall safety, as mRNA is not incorporated into the genome, to name a few [60]. Because CF is a single-gene based disease, this model of treatment is highly promising for a potential gene therapy, as shown by an ongoing clinical trial testing the delivery of wtCFTR-mRNA via inhalation of lipid nanoparticles [3].

We have been using wtCFTR-mRNA to transfect CF airway epithelial cells to correct CFTR function and subsequently abnormal chloride secretion constantly optimizing the transfection procedure [10,12]. In this study, the CF cell line CFBE41o- was transfected with wtCFTR-mRNA, using blank and capsaicin-loaded CS-lecithin oil-core nanocapsules. Functional Ussing chamber measurements were conducted 24 h after transfection and revealed significantly increased cAMP-dependent short-circuit current to the levels of healthy control cells in cells transfected with the capsaicin-loaded nanocapsules (Figure 8b). Furthermore, cells were transfected with ΔF508-CFTR-mRNA as negative control, followed by Ussing chamber measurements revealing no change in cAMP response (Figure 8b). These results indicate successful wtCFTR-mRNA transfection and, hence, increased CFTR function.

Approximately 5% of normal wtCFTR-mRNA level is sufficient to avoid the severe complications of CF [61]. For a sufficient chloride secretion, the airway epithelium requires about 6–10% epithelial cells expressing functional CFTR [62]. Mucociliary clearance is guaranteed with 25% epithelial cells expressing the chloride channel [63]. In a previous study, we were able to determine a transfection efficiency over 50%, using CS in the same cell line [13]. Therefore, and because of the restored cAMP-dependent short-circuit current shown with the Ussing chamber measurements, it can be assumed that in this study enough cells were transfected and hence express functional CFTR for sufficient chloride efflux and further regulatory functions.

The delivery of chemically modified wtCFTR-mRNA using nanomedicine in vivo was recently reported demonstrating the relevance of nanotechnology in CF research. Haque et al. used CS-coated poly-(lactic-co-glycolic acid) nanoparticles assembled with CFTR-mRNA for intratracheal spraying and intravenous injection in CFTR deficient (CFTR^tm1Unc^) mice. They reported significantly improved lung function in the tested rodents [64]. Robinson et al. were able to restore CFTR-mediated chloride secretion for at least two weeks and recovered around 55% of the net chloride efflux in the same mouse line. To achieve this outcome, they used lipid-based nanoparticles for nasal application [65]. These studies corroborate the results of the Ussing chamber experiments in this study, which also demonstrated increased CFTR function after wtCFTR-mRNA delivery, using nanoparticles. However, the murine model of CF often fails to reflect human pulmonary pathology [66,67]. Therefore, the pertinence of these studies should be regarded cautiously.

In contrast to the results achieved with capsaicin-loaded nanocapsules, cells transfected with the blank nanocapsules did not show any change in cAMP-dependent short-circuit current. The difference in transfection efficiency might be explained by the ability of capsaicin to reversibly open tight junctions [21], increasing the probability of the loaded nanocapsules to enter the cells. Furthermore, the MTT test revealed that blank nanocapsules displayed a higher cytotoxicity than the capsaicin-loaded nanocapsules (Figure 7), proposing that capsaicin might have a beneficial effect on nasal epithelial cells. Similar results were reported in a study of Kaiser and colleagues. They described significantly decreased cell viability of canine kidney epithelial cells (MDCK-C7) when incubated with blank nanocapsules, while capsaicin-loaded nanocapsules did not alter the viability of the cells [22]. The increased cytotoxicity might be the reason for the reduced transfection efficiency of blank nanocapsules.

In CF, the loss of CFTR function not only leads to decreased chloride secretion, but also to sodium hyperabsorption through the negatively regulated ENaC further thickening the mucus resulting in the classical clinical picture [68]. Therefore, the amiloride-sensitive current of CFBE41o- cells transfected with wtCFTR-mRNA was also assessed in this study, as amiloride is a specific blocker of ENaC. In contrast to the expectations, amiloride-sensitive current of cells was increased after wtCFTR-mRNA transfection, using nanocapsules (Figure 8c). CS has been reported to have various effects on proteins of the cell. For example, CS is able to translocate tight junction proteins from the cell membrane to the cytoskeleton [69]. Furthermore, it was shown that, after incubation with CS-sodium tripolyphosphate particles, liver cells upregulated eight proteins (Amyloid beta A4 protein, C-C chemokine receptor type 8, CD44 antigen, G2/mitotic-specific cyclin-B2, Neuroblast differentiation-associated protein, Osteonectin/SPARC protein, Glutamate receptor ionotropic, kainite 5, and Rho GTPase-activating protein 6) associated with regulative functions, such as differentiation, cell cycle regulation, or cell division [70]. Considering this, an effect of CS on ENaC might be the reason for the increased amiloride-sensitive current after application of the CS nanocapsules.

While the increase of amiloride-sensitive current was only minor in cells transfected with capsaicin-loaded nanocapsules, the increase after transfection with blank nanocapsules was significant. The observed difference in amiloride-sensitive current, albeit small, might be explained by the activation of TRPV1 through capsaicin in cells treated with the capsaicin-loaded nanocapsules. This theory is further corroborated by the results achieved after transfection with ΔF508-CFTR-mRNA. Even though no significant difference in amiloride-sensitive current could be observed compared to untreated control cells, the amiloride-sensitive current in cells treated with the capsaicin-loaded nanocapsules was slightly lower than in cells treated with the blank nanocapsules or the untreated cells (Figure 8c). While the activation of TRPV1 through its agonist capsaicin usually leads to acute and persistent pain [71,72], a study of Li and colleagues reported reduced α-ENaC expression and ENaC activity in the kidney of mice after TRPV1 activation via capsaicin [31]. According to them, ENaC activity is inhibited indirectly through protein kinase C, which is activated by TRPV1 [31], a theory, which is supported by another report showing inhibited ENaC currents in the lungs of rats after activation of protein kinase C [73].

However, the mice in the study of Li and colleagues were exposed to the vanilloid for 10 months, through a capsaicin diet, before the experiments were conducted [31]. Another study showing positive effects of capsaicin on nasal epithelial cells conducing in vivo studies with patients suffering from idiopathic rhinitis applying capsaicin to the nasal mucosa reported those effects after 12 weeks of treatment [74]. Therefore, 24 h of incubation with capsaicin-loaded nanocapsules, as conducted in the present study, might not be enough to evoke the full effect of the vanilloid.

The cell lines used in this study might be a good model for the investigation of chloride secretion; however, they do not represent the classical picture of CF regarding sodium hyperabsorption. CFBE41o- cells have been reported to display reduced ENaC expression and amiloride-sensitive current [75,76], as does the healthy equivalent cell line 16HBE14o- [77]. The dubiety of the two cells lines regarding ENaC activity was confirmed with functional Ussing chamber experiments, as both healthy and CF cell lines displayed the same amount of amiloride-sensitive current (Figure 8c). An adequate evaluation of the presented data regarding ENaC activity is, hence, not feasible. Therefore, *transcript therapy* alone might be sufficient to restore the imbalanced ion transports in CF in vivo.

## 5. Conclusions

In this study, we successfully developed natural, non-toxic, and non-viral nanocarriers for the delivery of mRNA to target cells. The nanocapsules, based on the natural aminopolysaccharide chitosan, displayed promising characteristics for transfection experiments, such as small size, high stability in transfection medium, and low cytotoxicity. We showed successful incorporation of capsaicin into the nanocapsules, as well as surface-loading of wtCFTR-mRNA, further increasing their suitability for transfection purposes. Finally, transepithelial measurements in a new state-of-the-art Ussing chamber demonstrated effectively restored CFTR function in the cystic fibrosis cell line CFBE41o- after transfection with wtCFTR-mRNA, using nanocapsules loaded with capsaicin. The results presented in this study reinforce the promising opportunities of nanobiotechnology not only for a possible cystic fibrosis *transcript therapy*, but for non-viral gene delivery in general.

## Figures and Tables

**Figure 1 biomedicines-08-00364-f001:**
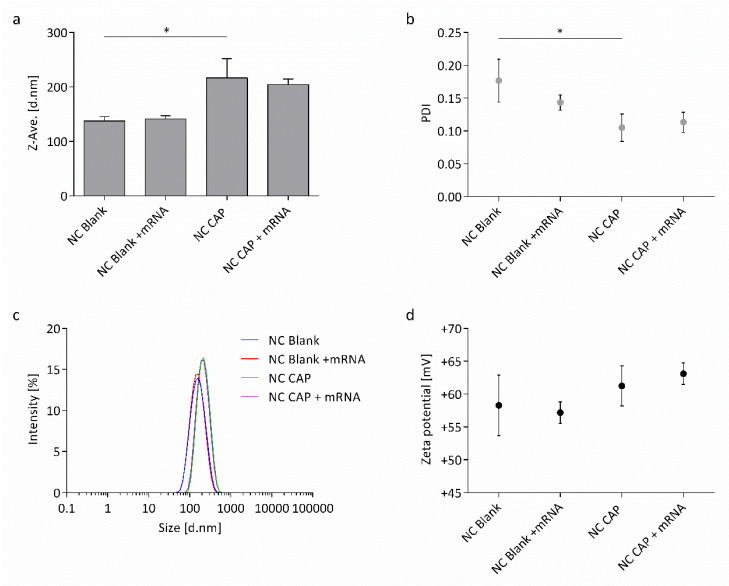
Physicochemical properties of chitosan (CS) nanocapsules. Shown is (**a**) the Z-average hydrodynamic diameter, (**b**) polydispersity index (PDI), (**c**) relative intensity of scattered light, and (**d**) zeta potential of blank (NC Blank) and capsaicin-loaded (NC CAP) nanocapsules either naked or surface-loaded with wtCFTR-mRNA at P/N charge ratio 75 (*p* ≤ 0.05 (*); *n* = 3).

**Figure 2 biomedicines-08-00364-f002:**
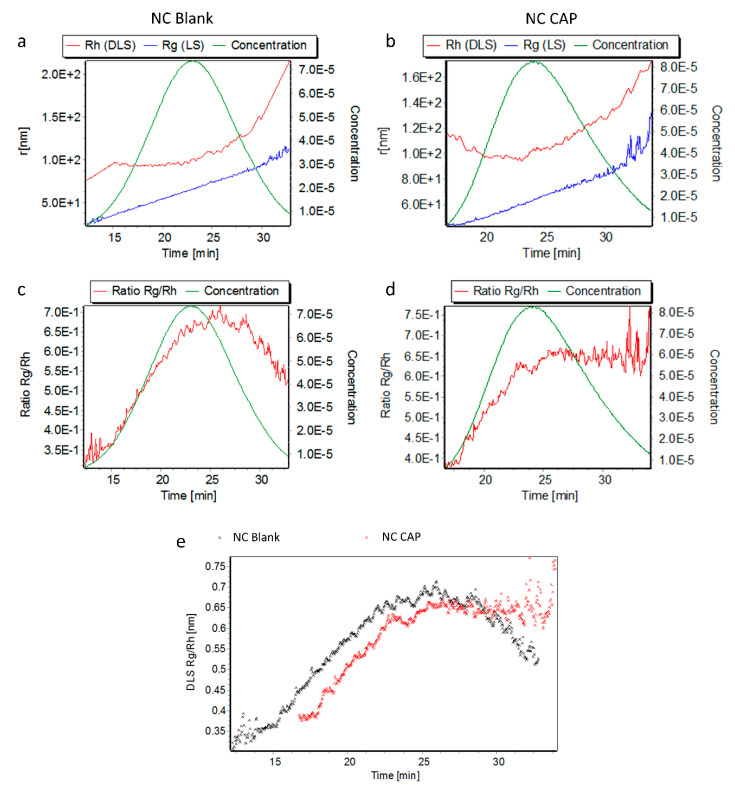
Asymmetric flow field-flow fractionation of blank (NC Blank) and capsaicin-loaded (NC CAP) CS nanocapsules. (**a**,**b**) Hydrodynamic radii (R_h_) determined by DLS-NIBS online (red line), gyration radii (R_g_) determined by MALS (blue line), and concentration (green line) over elution time. (**c**,**d**) R_g_/R_h_ ratio (red line) and concentration (green line) and (**e**) R_g_/R_h_ ratios of blank (black stars) and capsaicin-loaded (red stars) nanocapsules over elution time.

**Figure 3 biomedicines-08-00364-f003:**
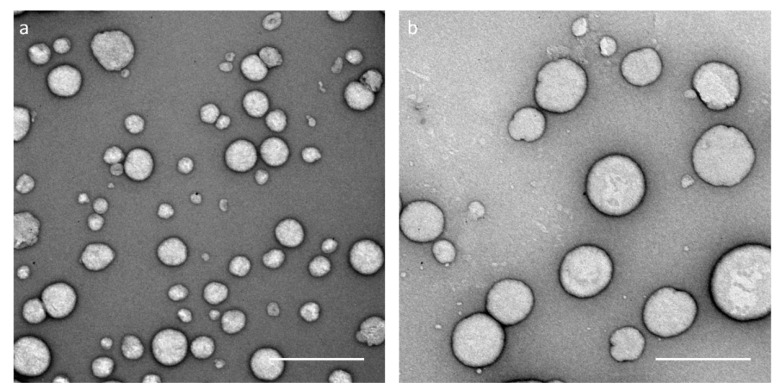
Representative transmission electron microscopy images of CS nanocapsules. (**a**) Blank nanocapsules and (**b**) capsaicin-loaded nanocapsules (uranyl acetate staining; scale bar = 500 nm).

**Figure 4 biomedicines-08-00364-f004:**
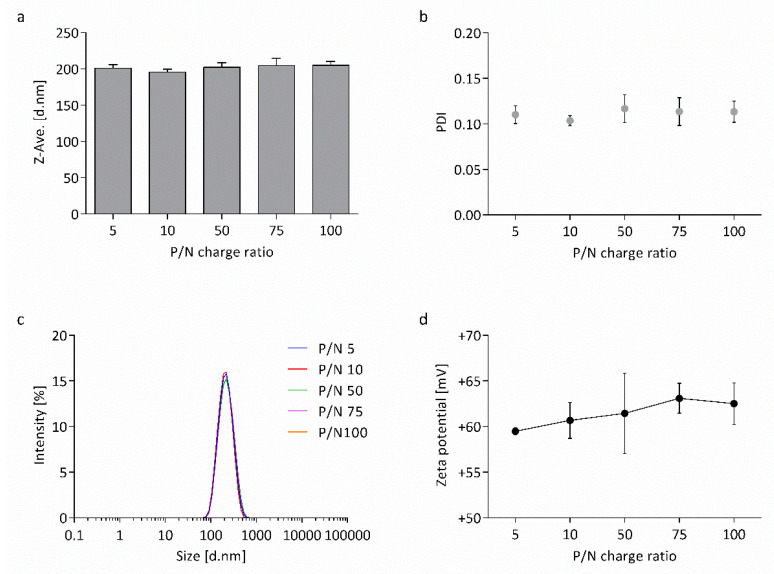
Physicochemical properties of CS nanocapsules surface-loaded with mRNA at varying P/N charge ratios. Shown is (**a**) the Z-average hydrodynamic diameter, (**b**) PDI, (**c**) relative intensity of scattered light, and (**d**) zeta potential of capsaicin-loaded nanocapsules surface-loaded with wtCFTR-mRNA (*n* = 3).

**Figure 5 biomedicines-08-00364-f005:**
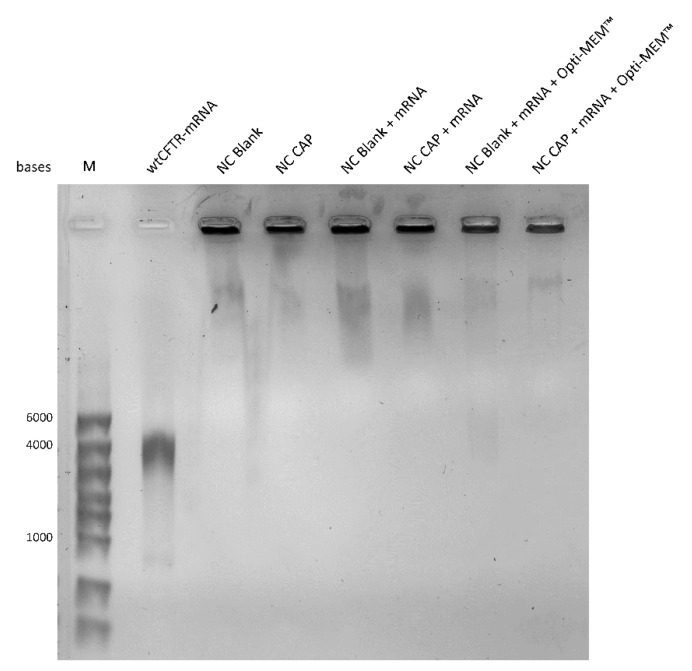
Gel retardation assay of CS nanocapsules surface-loaded with mRNA. Shown is the result of a 1% agarose-formaldehyde gel electrophoresis of blank (NC Blank) and capsaicin-loaded (NC CAP) nanocapsules either naked or surface-loaded with wtCFTR-mRNA at P/N charge ratio 75 in water and in transfection medium (Opti-MEM™) after 24 h incubation at 37 °C. Naked wtCFTR-mRNA showed a band at 4000 bases. Nanocapsules, both naked and surface-loaded, were retained in the pocket. Marker (M): RiboRuler™ High Range RNA Ladder (Thermo Fisher Scientific, Waltham, MA, USA).

**Figure 6 biomedicines-08-00364-f006:**
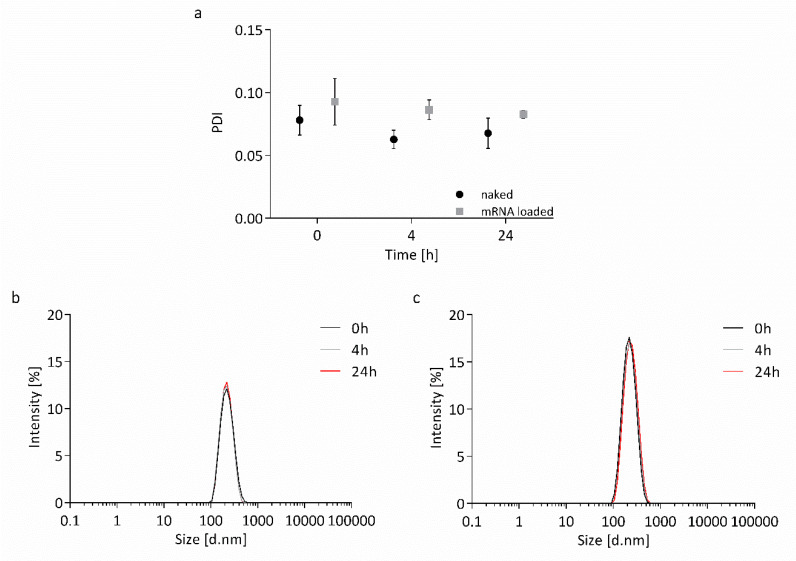
Size distribution by intensity of CS nanocapsules in transfection medium. Shown is the stability of capsaicin-loaded nanocapsules as (**a**) PDI and either (**b**) naked or (**c**) surface-loaded with wtCFTR-mRNA at P/N charge ratio 75 as relative intensity of scattered light. Nanocapsules were incubated in Opti-MEM™ supplemented with HEPES (20 mM) and mannitol (270 mM) at 37 °C. Both naked and wtCFTR-mRNA-loaded nanocapsules remained stable over time (*n* = 3).

**Figure 7 biomedicines-08-00364-f007:**
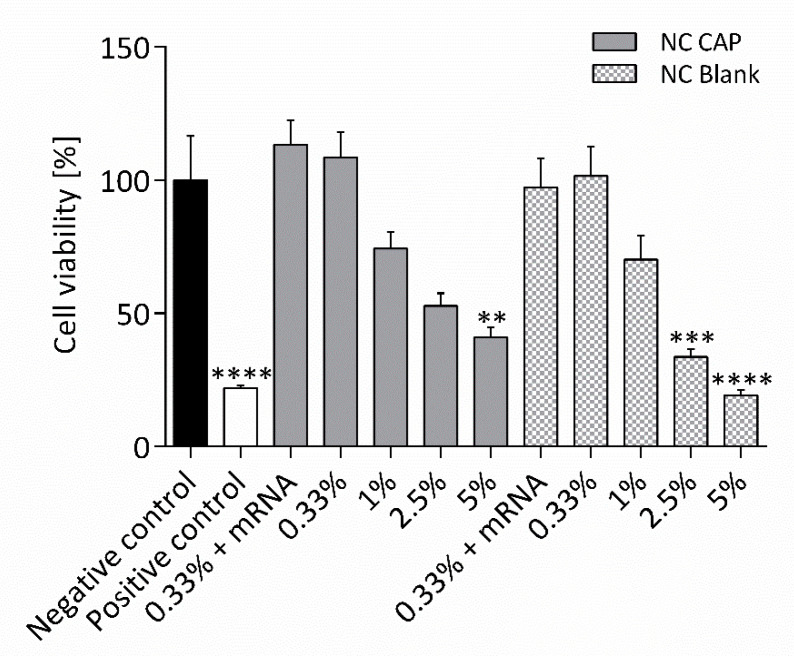
Effect of CS nanocapsules on the viability of CFBE41o- cells. Cells were incubated with blank nanocapsules (NC Blank) and nanocapsules loaded with capsaicin (NC CAP), at varying concentrations in Opti-MEM™, for 24 h, before an MTT assay was conducted. Concentrations correspond to 33, 100, 250, and 500 µM capsaicin in cells incubated with NC CAP. Cell culture medium was used as negative control; Triton^®^ X-100 was used as positive control. Significance was calculated compared to the negative control (*p* ≤ 0.01 (**), *p* ≤ 0.001 (***), and *p* ≤ 0.0001 (****); *n* = 3).

**Figure 8 biomedicines-08-00364-f008:**
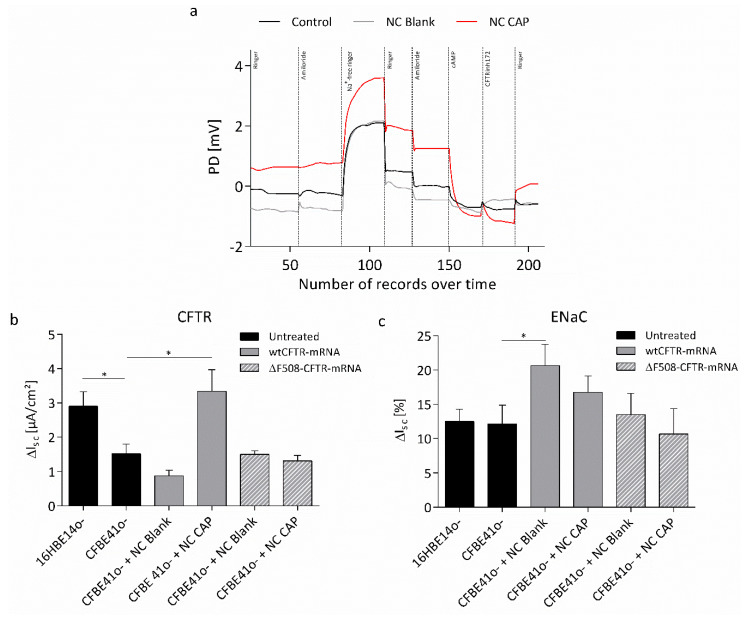
Transepithelial Ussing chamber measurements of epithelial cells before and after transfection with mRNA. (**a**) Potential difference (PD) of representative time courses of non-transfected CFBE41o- cells (control) and CFBE41o- cells transfected with wtCFTR-mRNA, using blank nanocapsules (NC Blank) and capsaicin-loaded nanocapsules (NC CAP), respectively. (**b**) Statistical evaluation of cAMP-dependent short-circuit current (I_sc_). Shown is the change of I_sc_ after CFTR activation by application of cAMP. (**c**) Statistical evaluation of amiloride-sensitive I_sc_. Shown is the percentage of sodium-current mediated by ENaC. Cells were transfected with 2.4 μg/cm^2^ wtCFTR-mRNA or ΔF508-CFTR-mRNA, respectively. Control cells were not transfected. Measurements were conducted 24 h after transfection (*p* ≤ 0.05 (*); 16HBE14o- *n* = 24; CFBE41o- *n* = 24; CFBE41o- + NC Blank (wtCFTR-mRNA) *n* = 20; CFBE41o- + NC CAP (wtCFTR-mRNA) *n* = 28; CFBE41o- + NC Blank (ΔF508-CFTR-mRNA) *n* = 24; CFBE41o- + NC CAP (ΔF508-CFTR-mRNA) *n* = 24).

**Table 1 biomedicines-08-00364-t001:** Composition of the chitosan (CS) nanocapsules surface-loaded with wtCFTR-mRNA, at varying positive/negative (P/N) charge ratios.

Charge Ratio	wtCFTR-mRNA		Chitosan	
P/N ^1^	(nmol) ^2^	(µg/µL)	(nmol) ^3^	(µg/µL)
5	33.8	0.053	185.1	2.5
10	17.8	0.053	185.1	2.5
50	3.7	0.053	185.1	2.5
75	2.5	0.053	185.1	2.5
100	1.9	0.053	185.1	2.5

^1^ Charge ratio (P/N): molar ratio of equivalent charges of NH_3_^+^/PO_4_^−^, ^2^ wtCFTR-mRNA (nmol): equivalent concentration of PO_4_^−^ from the wtCFTR-mRNA, ^3^ Chitosan (nmol): equivalent concentration of NH_3_^+^ from CS.

**Table 2 biomedicines-08-00364-t002:** Size distribution of CS nanocapsules determined by asymmetric flow field-flow fractionation. Average gyration radii (R_g_) determined by MALS, hydrodynamic radii (R_h_) determined by DLS-NIBS and shape factor (*ρ* = R_g_/R_h_) of blank (NC Blank) and capsaicin-loaded (NC CAP) nanocapsules (*n* = 3).

Nanocapsules	R_g_ (nm)	R_h_ (nm)	*ρ* = R_g_/R_h_
NC Blank	72.70 ± 1.34	103.33 ± 3.12	0.70 ± 0.01
NC CAP	71.03 ± 0.05	102.50 ± 6.38	0.69 ± 0.05

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
