# Peer review of "Capsaicin-Loaded Chitosan Nanocapsules for wtCFTR-mRNA Delivery to a Cystic Fibrosis Cell Line"

_biomedicines, 2020, doi:10.3390/biomedicines8090364_

Round 1

Reviewer 1 Report

Please determine the effect of capsaicin nanoparticles on the viability of normal bronchial epithelial cells. This is important because this will show whether the nanoparticles are cytotoxic to normal lung cells.

The experiments with transfection of CFTRmRNA lack an important control; that of a transfection of control mRNA (this would like a non-targeting siRNA which needs to be there for all experiments) so that the observed changes are not transfection artifacts 

Reviewer 2 Report

General comments:

- Kolonko A.K. et al. propose a very innovative study with a high translational power to the clinical field of CF. The description of the different methods and results is well clear. Minor comments/suggestions are reported below.

Specific comments:

- The authors do not mention in the text the supplementary figure attached to the manuscript. Please adjust.

- Line 51: Some references should be included, about “fat maldigestion due to pancreatic insufficiency” in CF patients, such as, Gelzo M. et al. Reduced absorption and enhanced synthesis of cholesterol in patients with cystic fibrosis: a preliminary study of plasma sterols. Clin Chem Lab Med. 2016; Singh V.K., Schwarzenberg S.J. Pancreatic insufficiency in Cystic Fibrosis. J Cyst Fibros. 2017.

- Line 55: Recent studies aimed to a personalized medicine for CF patients should be also cited (e.g., Di Lullo AM et al. An “ex vivo model” contributing to the diagnosis and evaluation of new drugs in cystic fibrosis. Acta Otorhinolaryngologica Italica 2017).

- Line 267: On the basis of which criterion did the authors choose to evaluate the physicochemical characteristics of the nanocapsules with P/N ratio of 75? Please, explain.

- Line 283: I suggest moving Table 2 into supplementary files as it appears redundant with the results in Figure 1.

Author Response

Answer to the referee

We are thankful for the constructive criticism and suggestions of the referee that helped us to improve the present version of our manuscript, which we thoroughlyrevised as the referee suggested. We addressed the following issues:

  1. Supplementary files

We supplied the original image of the gel retardation assay in the supplementary material. The referee asked to refer to the figure in the text. We therefore added the following sentence to the results section (line 348-349):

“The original image of the gel retardation assay can be found in the supplementary material (FigureS1).”

Furthermore, the referee suggested moving the table summarizing the physicochemical properties of the nanocapsules into the supplementary material as it appears redundant with the results presented in Figure 1. We moved the table to the supplementary material and added the following sentence to the results section (line 282-284):

The physicochemical properties of the nanocapsules are summarized in Table S1 in the supplementary material.

2.References

In the introduction section, we give a short overview about the clinical picture of cystic fibrosis (CF) as well as therapy approaches towards improvement of the patients’ quality of life. The referee suggested including some references about fat maldigestion due to pancreatic insufficiency and recent studies aimed to a personalized medicine for CF patients. We therefore added the following references to our manuscript:

Line 53:Gelzo, M.; Sica, C.; Elce, A.; Dello Russo, A.; Iacotucci, P.; Carnovale, V.; Raia, V.; Salvatore, D.; Corso, G.; Castaldo, G. Reduced absorption and enhanced synthesis of cholesterol in patients with cystic fibrosis: A preliminary study of plasma sterols.Clin. Chem. Lab. Med.2016, 54, 14611466.Singh, V.K.; Schwarzenberg, S.J. Pancreatic insufficiency in Cystic Fibrosis. J. Cyst. Fibros.2017, 16, S70S78.

Line 58:Di Lullo, A.M.; Scorza, M.; Amato, F.; Comegna, M.; Raia, V.; Maiuri, L.; Ilardi, G.; Cantone, E.; Castaldo, G.; Iengo, M. An "ex vivo model” contributing to the diagnosis and evaluation of new drugs in cystic fibrosis. Acta Otorhinolaryngol. Ital.2017, 37, 207213.

3.Charge ratio of the nanocapsules

During the initial physicochemical characterization of the nanocapsules, we decided to test the effect of loading the nanocapsules with wtCFTR-mRNA at a positive/negative (P/N) charge ratio of 75. The referee asked to provide a reason for this decision. We therefore added the following sentence to the results section (lines 272-274):

This specific P/N charge ratio was chosen based on the transfection procedureas atP/N75 the amount of wtCFTR-mRNA aswell as capsaicin are ideal for transfection of CFBE41o-cells.”

Reviewer 3 Report

The research reported is an extension of the experiments of the research team into a new direction with a new hypothesis.The topic also comes under the scope of the journal. However, few revisions are required. The article seems well articulated with standard English, but more informal sentences have been used. However, the following recommendations are advised in terms of new experiments and to give additional justification by the authors for further assessing its suitability for peer-review. However, the following major recommendations are required to be addressed by the authors.

1.      Abstract should be concise, highlighting the significant findings of the research. The introduction about the CF needs to be trimmed.

2.      The abstract conclusion sounds vague, authors need to rationalize and write what and how the results or designed delivery system can be useful or advantageous in mRNA delivery. Authors are advised to make suitable corrections.

3.      Line 72 the term adsorbed needs to be corrected.

4.      The introduction does not necessarily end with specific objectives, for example, prepared either blank nanocapsules and nanocapsules loaded with capsaicin? This para should be we written properly. Also, the authors need to eliminate discussing results in this para-repeated the characterization showing no change to the nanocapsules’ attributes after adsorption of the nucleic acid.  The informal writing should be minimized, and tense should be corrected.

5.      Line 107, statement “Finally, the mRNA was precipitated with ethanol/ammonium acetate to achieve” contradicts, please rephrase and write meaningful.

6.      Majority of metholodgies sections lack citation of references from where the procedures were adopted. It is advised to add the needful.

7.      Authors did not disclose how the formulation parameters were optimized and fixed for respective quantities or concentrations as optimized for preparing chitosan nanocapsules. Authors are advised to justify this and also why optimization tools such as Box-Behnken design have not been used in the study.

8.      The authors right from the beginning quote “coated” or coating of cap nanoshells with mRNA which is wrong and not scientific. Chitosan is cationic and is expected to form a “COMPLEX” with the anionic mRNA. Essentially does not get coated or adsorbed onto the shell surface but forms a network. Authors need to justify this error and change the needful through out the manuscript. Strongly recommended.

9.      The stability measurement produced or used sounds not correct protocol.  Starting maintenance temperature of the complex is not listed. The procedure needs to mention that PDI is being considered as a parameter to measure the stability. Number of measurements needs to be mentioned as well. Strongly recommended.

10.   The authors should have even performed gel retardation assay post the measurements time of the stability assessments. This could have given a reasonable data indicating the real complexation efficiency of the prepared delivery system with mRNA. Authors are advised for these experiments and include the data. Strongly recommended.

11.  It is unclear why the cell viability of nanoshells complexed with  mRNA is performed only for one time point or for 24 hr? why it is not checked for Day 3 or Day 7? The authors need to justify or else it would raise concerns over the stability of the developed delivery system. Strongly recommended.

12.  Authors need to justify or rewrite “nanocapsules were added to the apical side” line 223.

13.  Line 269 “increased significantly after loading of the alkaloid” is meaningless- kindly rephrase meaningfully.

14.  Authors need to justify how PDI decreases when nanoshells get complexed with mRNA and increase their size that can actually lead to agglomeration and further increase the size and effect the stability of the delivery system. Authors need to justify and write additional information for the same. Line 272-277. This is very contradicting.

15.  It is surprising to see that NC-CAP coated is exhibiting narrow and small size in comparison to the uncoated NC CAP. Authors need to check their results and figures presented.

16.  Authors have not mentioned why “The nucleic acid affected neither the Z-average hydrodynamic diameter, nor the PDI of the nanocapsules”. Line 320-321.

17.  From figure it is evident that Figure 7 NC CAP and NC blank  at 2.5 and 5% have greatly reduced the cell viability but authors report it differently in line 370-371. Rewrite correctly and meaningfully.

18.  Correct spelling “using” Line 383.

19.  Conclusion needs to be revised highlighting the importance and key findings of the study.

Round 2

Reviewer 3 Report

Authors have carefully  addressed the recommendations but are also advised for the following,

- Cite relevant chitosan-based carrier systems references as below listed in the introductory, optimization, and characterization sections explaining the reported advantages and importance of chitosan in designing suitable carrier based systems.

 https://doi.org/10.3390/pharmaceutics12070652

https://doi.org/10.1016/j.matpr.2020.01.491

https://doi.org/10.3390/md18040226

Author Response

Answer to the referee

The referee suggested including chitosan-based carrier systems references to explainthe advantages and importance of chitosan in designing suitable carrier based systems.

We appreciated the suggestions and tried to improve the manuscript according to the helpful recommendations.We included the following sentences to our manuscript together with the appropriate literature:

Lines 66-68:

Because of its positive attributes such as biocompatibility, mucoadhesiveness, low cytotoxicity and low immunogenicity, CS has already been used as a carrier for various drugs in multiple studies [13].”

Lines: 446-453:

Features such as physical and chemical properties, as well as biological performance (e.g. transfection efficiency) of CS, are dependent on molecular weight and degree of acetylation of the polysaccharide [4]. Therefore, it iscrucial to rigorously characterize CS in order to design the most optimal formulation for a specific purpose. To further optimize the suitability of CS-based nanocarriersas delivery systems, it is essential to characterize them for their physicochemical properties and understand them on a molecular level [5,6].Thus, we characterized the nanocapsules regarding their physicochemical properties.

References cited in this letter:

  1. Alhakamy, N.A.; Fahmy, U.A.; Ahmed, O.A.A.; Caruso, G.; Caraci, F.; Asfour, H.Z.; Bakhrebah, M.A.; N. Alomary, M.; Abdulaal, W.H.; Okbazghi, S.Z.; et al. Chitosan coated microparticles enhance simvastatin colon targeting and pro-apoptotic activity. Mar. Drugs2020, 18, 226.
  2. Alhakamy, N.A.; Ahmed, O.A.A.; Kurakula, M.; Caruso, G.; Caraci, F.; Asfour, H.Z.; Alfarsi, A.; Eid, B.G.; Mohamed, A.I.; K. Alruwaili, N.; et al. Chitosan-based microparticles enhance ellagic acid’s colon targeting and proapoptotic activity. Pharmaceutics2020, 12, 652.
  3. Fernández Fernández, E.; Santos-Carballal, B.; de Santi, C.; Ramsey, J.; MacLoughlin, R.; Cryan, S.-A.; Greene, C. Biopolymer-based nanoparticles for cystic fibrosis lung gene therapy studies. Materials (Basel).2018, 11, 122.
  4. Mao, S.; Sun, W.; Kissel, T. Chitosan-based formulations for delivery of DNA and siRNA. Adv. Drug Deliv. Rev.2010, 62, 1227.
  5. Raghavendra Naveen, N.; Kurakula, M.; Gowthami, B. Process optimization by response surface methodology forpreparation and evaluation of methotrexate loaded chitosan nanoparticles. Mater. Today Proc.2020.
  6. Santos-Carballal, B.; Fernández, E.F.; Goycoolea, F.M. Chitosan in non-viral gene delivery: role of structure, characterization methods, and insights incancer and rare diseases therapies. Polymers (Basel).2018, 10, 151.